# Identifying Potential Molecular Targets in Fungi Based on (Dis)Similarities in Binding Site Architecture with Proteins of the Human Pharmacolome

**DOI:** 10.3390/molecules28020692

**Published:** 2023-01-10

**Authors:** Johann E. Bedoya-Cardona, Marcela Rubio-Carrasquilla, Iliana M. Ramírez-Velásquez, Mario S. Valdés-Tresanco, Ernesto Moreno

**Affiliations:** 1Facultad de Ciencias Básicas, Universidad de Medellín, Medellin 050026, Colombia; 2Corporación para Investigaciones Biológicas, Medellin 050034, Colombia; 3Instituto Tecnológico Metropolitano, Medellin 050034, Colombia

**Keywords:** fungal pathogens, drug repurposing, drug development, new therapeutic targets, structural bioinformatics, MEK inhibitors

## Abstract

Invasive fungal infections represent a public health problem that worsens over the years with the increasing resistance to current antimycotic agents. Therefore, there is a compelling medical need of widening the antifungal drug repertoire, following different methods such as drug repositioning, identification and validation of new molecular targets and developing new inhibitors against these targets. In this work we developed a structure-based strategy for drug repositioning and new drug design, which can be applied to infectious fungi and other pathogens. Instead of applying the commonly accepted off-target criterion to discard fungal proteins with close homologues in humans, the core of our approach consists in identifying fungal proteins with active sites that are structurally similar, but preferably not identical to binding sites of proteins from the so-called “human pharmacolome”. Using structural information from thousands of human protein target-inhibitor complexes, we identified dozens of proteins in fungal species of the genera *Histoplasma*, *Candida*, *Cryptococcus*, *Aspergillus* and *Fusarium*, which might be exploited for drug repositioning and, more importantly, also for the design of new fungus-specific inhibitors. As a case study, we present the in vitro experiments performed with a set of selected inhibitors of the human mitogen-activated protein kinases 1/2 (MEK1/2), several of which showed a marked cytotoxic activity in different fungal species.

## 1. Introduction

Invasive fungal infections (IFIs), caused by yeasts and filamentous fungi, are opportunistic infections that occur mostly in immunodepressed patients and in patients in critical conditions, causing a high morbidity and mortality [1]. IFIs may manifest with different intensities, from simple and mild infections, as is the case of external mycoses, to severe systemic and disseminated mycoses that can cause death [2]. The epidemiological landscape of invasive mycoses is in continuous change, driven by etiological variations among hospitals, countries and the influence of multiple local variables, patient risk factors and medical and surgical praxis [3].

The current repertoire of antifungal drugs includes different classes of molecules: pyrimidines, polyenes, echinocandins and azoles [4,5,6]. These antifungal drugs, however, present several important drawbacks, such as their adverse side effects, the increasing resistance developed by many fungal pathogens and long treatment times [7]. Therefore, there is a compelling medical need for broadening the therapeutic alternatives to treat these infections. Two main alternatives in this route are drug repositioning and the development of new drugs directed to new molecular targets in fungi. In either case, a favorable balance between clinical benefits and adverse effects is a relevant issue to take into account.

Commonly, the identification of new targets in pathogens focuses on unique proteins, not present in humans, or with low sequence similarity with human proteins. This approach intends to minimize possible cross-reactions leading to adverse secondary effects. For example, Sosa et al. (2018) developed a database (Target-Pathogen) and a search system that integrates multiple sources of information for the identification of possible targets in pathogens [8]. Among the filters applied in this search system is the so-called “off-target criterion”, which discards proteins with close homologues in humans. Similarly, in more recent works, Mukherjee and coworkers applied a “subtractive genomics” approach to filter out homologous proteins in the search for targets in *Candida* species [9], while Palumbo and coworkers applied the same off-target criterion in a search for potential targets in *Listeria monocytogenes*, following a multilayer omics strategy [10].

Drug repositioning, also referred to as drug repurposing, is a different strategy consisting in finding new medical uses for approved drugs or compounds that have shown an acceptable safety profile in clinical trials, including those that have failed in later stages during the development. This strategy implies shorter development times, lower costs and fewer risks [11,12]. Successful drug repurposing cases have been reported in various therapeutic areas, prompting pharmaceutical companies to open up collaborations with biotech firms and academic communities to synergize research in this area [12,13,14].

The emergence of various genomic, drug and disease knowledge databases has promoted the rapid development of a variety of computational approaches to guide drug repositioning and new drug development projects. Thus, the so-called network-based methods combine and exploit various kinds of information from multiple data sources, e.g., transcriptomics, drug-induced expression profiling, disease–disease associations, drug–drug interactions, among others [13,15]. Structure-based methods, on the other hand, rely on techniques such as protein–ligand docking, molecular dynamics simulations, virtual screening and quantitative structure–activity relationship (QSAR). Additionally, in recent years, the use of artificial intelligence methods in drug development is gaining a big momentum [15,16].

Drug repurposing approaches have been applied as well to fungal infections [17]. For example, finasteride, a drug generally used for the treatment of benign prostatic hyperplasia, showed efficacy in the prevention of biofilm formation by *Candida albicans*, when used alone and in combination with fluconazole, and showed an effect also in the treatment of preformed biofilms [18]. Another example is atorvastatin, a drug used as a plasma cholesterol reducer, which showed antifungal activity by inhibiting the production of ergosterol from the cell wall in five *Candida* strains (*C. albicans*, *C. glabrata*, *C. kefyr*, *C. stellatoidea* and *C. krusei*) [19]. Furthermore, several drug libraries have been screened against different pathogenic fungi in the search of drugs with previously unknown antifungal effects [20,21,22,23,24,25,26].

Currently, the more than one and a half thousand FDA-approved drugs and several thousand compounds in clinical trials, together with their molecular targets, constitute a rich repertoire for drug repurposing and new drug development. According to a study published by Santos et al. (2017), by the year 2015 the Food and Drug Administration (FDA) in the United States had approved a total of 1578 drugs, targeting 893 different human and pathogen-derived biomolecules. This set of targets is defined in the article as the “pharmacolome”, which is spread across the fourteen groups of the anatomical therapeutic chemical (ATC) classification system. A quick survey over the compiled pharmacolome shows the limited availability of approved drugs to treat invasive fungal infections [27].

In this work we developed a structural bioinformatics strategy to identify potential therapeutic targets in fungi, test them in vitro using known drugs and inhibitors and, in suitable cases, intend to develop new fungus-specific inhibitors. The core of this approach consists in identifying fungal proteins with active sites that are structurally similar, but preferably not identical to binding sites of proteins from the human pharmacolome. On the one hand, a high structural similarity with a human counterpart allows validation of the fungal target using cross-reactive inhibitors of the human protein (possibly leading to drug repurposing). On the other hand, a few amino acid differences in the binding pocket would produce local topological and chemical changes that might be exploited for the design of new specific inhibitors of the fungal target.

Using structural information, we have identified dozens of proteins in several fungal species of the genera *Histoplasma*, *Candida*, *Cryptococcus*, *Aspergillus* and *Fusarium*, which might be exploited for drug repurposing and for the design of new antifungal agents. As case study we analyze a few fungal proteins showing binding sites similar to the non-ATP competitive binding site of the human mitogen-activated protein kinases 1 and 2 (MEK1/2), and present the in vitro experiments performed with a set of selected (MEK1/2) inhibitors, several of which showed marked cytotoxic activity in various fungal species. Importantly, the binding sites of the MEK analogs in several fungal species show mutations that create opportunities for the design of fungus-specific inhibitors.

## 2. Results and Discussion

### 2.1. Selected Set of Human Protein Targets and Binding Site Definition

The primary data source for this work was the list compiled by Santos and coworkers (2017), which included 549 protein targets of small drugs approved by the FDA up to 2015 [27]. We complemented these data by including the small drugs (and their protein targets) approved between 2016 and 2020, which added another 90 small drugs, resulting in a total of 639 human protein targets. From this set, 433 proteins included in their UniProt records cross-referenced to PDB structures, which amounted to more than 8500 PDB entries. The automated and subsequent manual analysis of all these structures, as described in Section 3, yielded 264 different protein targets in complex with one or more ligands.

Figure 1 shows the distribution of the number of PDB complexes per protein target. Most of the targets are represented in the PDB by more than a single protein–ligand complex, which allows a more comprehensive definition of the binding site. An extreme case is the estrogen receptor (UniProt ID: P03372), with more than 500 protein–ligand structures. In spite of this disparity in the numbers of complexes per target, we found consistent binding pocket definitions for most of the proteins. For example, for the estrogen receptor the only found binding site region, located between sequence positions 342 and 544, corresponds to the estradiol binding site.

The obtained set of PDB entries included 86 ligands corresponding to FDA-approved small drugs (Appendix A), which were distributed across > 400 complexes. The large majority of these drugs have >60% of their surface area buried in the protein upon complexation (Figure 2A), while the few cases showing a lower percent of buried area (for example, for cholic acid) corresponds to extra copies of the ligand lying on external areas of the protein surface. We decided to use this value of 60% of buried ligand surface, covering most of the complexes, as cutoff for further analysis of the binding sites. Likewise, we applied a molecular weight cutoff, allowing a maximum of 80 heavy atoms (corresponding roughly to 1.1 kDa), to discard large ligands, which were mostly peptides and oligonucleotides (see Figure 2B).

The analysis performed to delimit the binding regions within the protein sequences yielded around 1200 clusters of sequence regions, corresponding to 272 protein targets. By manually reviewing these clusters, we selected 343 binding regions in a total of 264 proteins from the human pharmacolome. About 30% of these proteins contained more than one pocket region.

Figure 2C shows the distribution of the number of amino acid residues per binding pocket, as defined here following a contact distance criterion. This means that each amino acid belonging to a binding pocket has at least one atom within a contact distance (4.5 Å) from a ligand in at least one PDB complex. These contacts include mostly amino acid side chains, but also residues that interact only through their backbone atoms. The numbers of pocket amino acids across different targets span a wide range, having a maximum at around 20–30 residues. These residues are distributed along sequence regions of different lengths, mostly within a range of 100–250 residues (Figure 2D). The largest regions correspond to transmembrane proteins, such as the alpha units of the sodium channel proteins 2, 9 and 4 (Q99250, Q15858, P35499) and the Voltage-dependent T-type calcium channel subunit alpha-1G (O43497), where the protein chain crosses the cell membrane several times, with large sequence stretches separating the ligand-binding segments.

### 2.2. Searching a Fungal Proteome for Binding Sites–Case Study: Histoplasma capsulatum

Here we present the results obtained for the *Histoplasma capsulatum* proteome as example of the application of the developed strategy. Figure 3A shows the significant differences between the results obtained using the full human protein sequences for BLAST and those obtained using the defined 343 binding site regions, even though the restrictions imposed for the second type of search were stronger: ≥80% sequence coverage vs. ≥40% for the full sequences (for most of the proteins, the binding region covers around 40–50% of the full sequence). As shown in Figure 3A, BLAST with binding regions yielded a significantly higher number of hits.

The similarity further increases when comparing only the sets of amino acids forming the binding pockets (Figure 3B), which for the fungal proteins were defined from their alignments with the human binding region sequences, as explained in Methods (Section 3). Even for proteins with low similarity (<30% aa identity) in their binding region sequences, the identity between the binding pocket amino acids may be considerably high. For example, the alignment for the aromatic-L-amino-acid decarboxylase (P20711, sequence region 147–303) yields a 33% aa identity with a sequence segment of a fungal protein (UniProt identifier C0NW51; annotated as a glutamate decarboxylase-like protein), while the identity of the corresponding binding pocket residues reaches 85%. Not surprisingly, highly similar binding pockets belong to proteins with conserved roles in the cell, as is the case of polymerases and other enzymes. Several of these binding pockets correspond to binding sites for ATP and different cofactors.

### 2.3. Expanding the Search to Other Fungal Proteomes

The above analysis carried out for *Histoplasma capsulatum* was extended to other five fungal proteomes of microorganisms of medical relevance: *Aspergillus fumigatus*, *Candida albicans*, *Candida parapsilosis*, *Cryptococcus neoformans* and *Fusarium oxysporum*. The main results from these analyses are summarized in Table 1, while the full list of hits is presented in Appendix A. The fungal proteins listed in Table 1 contain binding pockets showing ≥70% aa identity with their human counterparts. Interestingly, four of the human targets have orthologs with 100% conserved binding sites in all or most of the investigated fungal species.

It is worth noting that Table 1 shows, for each human target, only the highest ranked fungal protein. However, for several human targets we found two or three fungal proteins (within the same species) having similar binding pockets, with relatively small differences in their aa identity percentages. This is the case, for example, of the DNA polymerase delta catalytic subunit, which yielded two matches in each of the six proteomes. The binding region sequences of these fungal proteins differ in aa identity (38–60%) compared to the corresponding sequence region in the human target, but all of them contain very similar binding pockets (~90% aa identity). Appendix A shows the full lists of matches; additionally, see below as an example the results for MEK1/2 in Figure 4.

Several of the human proteins included in Table 1 are the targets of drugs and inhibitors that have been tested in fungi. For example, the cancer drug sorafenib, which targets multiple proteins, among them the P-glycoprotein 1 (P08183), was identified from a kinase inhibitor library screening as a strong inhibitor of *Histoplasma capsulatum* and *Cryptococcus neoformans* [28]. Statins such as atorvastatin and simvastatin, targeting the HMG-CoA reductase (P04035) have shown inhibitory effects in *Candida albicans*, *Candida Glabrata* and *Aspergillus fumigatus* [29]. Disulfiram, a drug inhibiting the aldehyde dehydrogenase (P05091) that is used to treat chronic alcoholism, showed strong inhibitory effects in *Candida albicans* and *Candida auris* [30]. The immunosuppressive drug tacrolimus, targeting the peptidyl-prolyl cis-trans isomerase FKBP1A (P62942) had effects in 11 fungi and 3 oomycetes of agricultural importance [31]. Finally, vorinostat, targeting histone deacetylases (Q92769, Q9UBN7) and used in the treatment of cutaneous T cell lymphomas, showed strong effects in *Aspergillus* spp. [32].

The identification in this work of fungal proteins with binding pockets similar to those of human proteins targeted by drugs that have shown inhibitory effects in fungi, not only serves as a strong support of the developed strategy, but also helps to identify the actual fungal targets and to understand the mechanisms of action of such drugs in these microorganisms. Furthermore, many of the human proteins included in Table 1 and Appendix A, are the targets of drugs and inhibitors that have not been tested yet in fungi, which opens up a large research space for drug repositioning and new drug development.

Since the fungal proteomes have been annotated mostly in an automated way, functional assignments for the identified proteins are not always reliable. Therefore, it would be difficult in many cases to establish direct functional relationships between the human targets and the identified fungal proteins having similar binding sites. For practical purposes, nonetheless, the obtained results lead straightforwardly to the use of known inhibitors of the human targets to test their effects in fungi. Such chemical probing of the predicted targets may be accomplished either by following a comprehensive in vitro testing of a large number of inhibitors (when available), or by following a computational modeling approach to define a more limited set of molecules to be tested, as we illustrate below with the in silico predictions and in vitro assays performed with inhibitors of the human MEK1/2 proteins.

### 2.4. Several MEK1/2 (MEK) Inhibitors Have Strong Inhibitory Effects in Various Pathogenic Fungi

In humans, the dual specificity mitogen-activated protein kinases 1 and 2 (MEK1 and MEK2, also known as MAP2K1 and MAP2K2), are essential components of the mitogen activated protein (MAP) kinase signal transduction pathway. Both MEK1 and MEK2 have a unique inhibitor-binding pocket adjacent to the Mg/ATP-binding site [33]. Currently, four MEK inhibitors have been approved by the FDA for cancer treatment: trametinib, binimetinib, selumetinib and cobimetinib [34] while others are in clinical trials. The web platform of Selleck Chemicals (Houston, TX, USA), for example, currently lists 33 commercially available MEK inhibitors.

In general, inhibitors of the PI3K/AKT/mTOR, RAS/RAF/MEK/ERK pathway, which are used in the treatment of malignancies and immune-mediated diseases, may predispose to fungal infections by suppressing important components of the adaptive and innate immune response [35], therefore, they would not likely be used as antifungal agents. Nonetheless, there are a few reports where MEK inhibitors have been tested in plant pathogenic fungi. For example, the MEK1/2 inhibitor U0126 was found to decrease germination and hyphae growth in *Aspergillus fumigatus* [36] and to inhibit the conidial germination and pathogenicity of *Setosphaeria turcica*, a plant pathogen [37].

The binding region sequence encompassing the non-ATP binding pocket in MEK1/2 goes from residue 78 to 219 (ca. 200 aa). In this region we identified 23 amino acids (identical in the two proteins) shaping the binding pocket inner surface. Running BLAST using the MEK1/2 binding region sequences yielded three proteins in each of the six analyzed proteomes, showing 62–77% of aa identity between their binding pocket residues and those of MEK (Figure 4).

The alignment in Figure 4 reveals a high degree of binding pocket conservation, with 10 out of 23 residues fully conserved across the human and all the fungal variants. Furthermore, in most cases the amino acid substitutions are conservative, as in positions 78, 99, 127, 141, 143, 212, 215 y 216. At positions 79 and 118, drastic substitutions (G/Y; L/G or L/A, respectively) appear in a few proteins in several fungal species. As discussed below, some of these substitutions represent interesting opportunities for the design of fungus-specific inhibitors.

We decided to test our predictions by assaying in vitro a set of reported MEK inhibitors on the six fungal species analyzed in silico. Docking simulations on the constructed models for proteins F0UAN5 and A0D2XNJ1 from *Histoplasma capsulatum* and *Fusarium oxysporum*, respectively, were performed for 25 inhibitors found in complex with MEK1 in the Protein Data Bank. As result, we selected seven inhibitors: cobimetinib [38], myricetin [39], refametinib [40], trametinib [41], GDC0623 [42], AZD6244 [43] and TAK-733 [44] for the in vitro assays.

Table 2 shows the results of the growth inhibition experiments performed for the six fungal species. The most susceptible microorganism was *Histoplasma capsulatum*, with four inhibitors (cobimetinib, GDC-0623, myricetin and refametinib) showing IC50 values in the low micromolar range. Similarly, *Aspergillus fumigatus* was strongly affected by three inhibitors (cobimetinib, GDC-0623 and TAK-733), while only one inhibitor (cobimetinib) showed a marked effect on *Fusarium oxysporum*. No inhibitor had effects on all the fungal species. The two tested Candida species were affected by two inhibitors each, but only at a high micromolar range (>100 μM). The use of a very low concentration of the SDS surfactant (0.002%), which most likely increases inhibitor solubility, improved the observed inhibitory effects in most cases. This concentration of SDS alone, or in combination with DMSO or ethanol, had only minor effects in fungal viability.

Since for each of the investigated fungal species we found three proteins with binding sites similar to that of the human MEKs, it is not possible to attribute the observed cytotoxic effects to a particular protein. Furthermore, and although less probable, the actual target might be a different, so far unidentified fungal protein. Reliable target validation would require complementary experiments, e.g., genetic manipulations to affect protein expression. In addition, as discussed below, target validation could be supported with growth inhibition assays involving compounds predicted to be specific for a particular fungal protein.

### 2.5. Opportunities for the Design of Fungus-Specific Inhibitors

Several of the fungal proteins in Figure 4 show amino acid substitutions in their binding pockets, as compared with the human MEKs, that cause small local topological changes, in particular mutations L118G (*A. fumigatus*, *H. capsulatum*) and L118A in the two *Candida* species. As illustrated in Figure 5 for the *Histoplasma capsulatum* protein F0UAN5, mutation L118G creates a void space within the binding site, previously occupied by the bulky Leu sidechain. This additional small cavity could be filled up by compounds with suitable chemical structures, which, on the other hand, would not bind to human MEK1/2 because of the steric hindrances caused by the leucine sidechain. As discussed above, the actual antifungal effect of these fungus-specific inhibitors would depend on the relevance of their targets for cell vitality.

Performing this kind of analysis on the different pairs of human and fungal proteins having similar binding pockets, as found in this study, may disclose many potential fungal targets with binding site mutations that open up a design space for fungus-specific inhibitors. The zone between 60–75% binding pocket aa identity (Figure 3B), which includes dozens of fungal proteins, looks particularly interesting in this regard.

## 3. Computational and Experimental Methods

### 3.1. Computational Strategy to Identify Potential Targets in Fungi and Other Pathogens

Our approach consists in identifying fungal proteins with active sites (meaning the set of residues lining the binding pocket) that are similar to active sites of proteins from the human pharmacolome. As mentioned in the Introduction, a high structural similarity with the binding site of a human counterpart facilitates a chemical validation of the fungal target using known inhibitors of the human protein and, ultimately, may lead to a drug repurposing strategy. We, however, are more focused on exploiting one or a few relevant amino acid differences in the binding pocket that would create a “design space” for new specific inhibitors of the fungal target.

Briefly, we employed a structural approach to identify binding site similarities, taking advantage of the thousands of available crystal structures for proteins of the human pharmacolome, many of them in complex with inhibitors. As explained in detail in the following sections, we used these bound inhibitors as anchors to define the binding site amino acids for each human target, followed by local sequence searches and analyses against the proteomes of several fungal species. The workflow is represented in Figure 6.

#### 3.1.1. Selection of the Human Protein Targets to Be Used for Fungal Proteome Searches

The list of FDA-approved small drugs and their protein targets, up to 2015 as compiled by Santos et al. (2017), was the main primary source for our work. We updated this list up to 2020 by including the small drugs approved by the FDA between 2016 and 2020, taken from the “Compilation of CDER NME and New Biologic Approvals 1985–2020” (www.fda.gov, accessed on 15 November 2021) and mapping their protein targets using the DrugBank database [45]. The compiled data included the generic drug names, their molecular weights, as well as the UniProt identifier [46] of their protein targets, which were used to retrieve the amino acid sequences and the available crystal structures that are associated with many of these proteins.

#### 3.1.2. Binding Site Definition at the Structural Level in the Human Targets

Binding site determination for a human target relied on the existence of at least one protein–ligand complex in the Protein Data Bank (PDB) [47]. Therefore, the next step was to determine which of the thousands of PDB structures associated with hundreds of human clinical protein targets contain bound inhibitors. For this purpose, we used our own program ‘complex_info’ [48], which identifies bound small ligands and carries out a detailed geometric analysis of the protein–ligand interactions, providing information on ligand size (number of heavy atoms), percent of buried ligand surface area, contacting protein atoms and amino acids, among other useful data. We used a filter of 10 heavy atoms as minimum to identify bound ligands, including small peptides and small nucleic acid chains.

Next, we focused the analysis on protein–ligand complexes containing FDA-approved drugs to gather statistics on the number of heavy atoms, surface area buried in the protein upon complexation and the number of contacting protein residues. We then used these data to adjust our search parameters and define more precisely the binding pockets in the human protein targets. In this process we excluded crystallographic molecules such as buffers and polyethylene glycols, heme groups and large peptides and nucleic acid ligands. Finally, for each obtained protein–ligand complex we defined the pocket region as the set of amino acid residues found within 4.5 Å from the ligand, using the VMD program [49]. For each of the identified complexes we tabulated the protein UniProt identifier, the PDB ligand ID, the number of ligand heavy atoms and the PDB sequence number of each binding pocket residue.

#### 3.1.3. Defining Binding Site Regions at the Sequence Level for the Human Targets

We reasoned that using the functionally conserved binding site regions of the human targets for a BLAST search would increase the chances of finding similar regions in fungal proteins. Therefore, the next step was to delimit, for each selected protein target, a continuous sequence region containing the binding site pocket, based on the list of individual binding site amino acids identified in the previous step. Commonly, these binding site residues were scattered along a large sequence segment of a few hundred amino acids. In many cases, more than one protein–ligand complex was available in the PDB for the same target, yielding slightly different binding site lists depending on the size and geometry of each ligand. In addition, the sequence numbering for the same protein may differ between PDB entries, which created an additional difficulty for mapping the binding site residues to the reference Uniprot sequence. To solve this problem, we used pentamer sequence segments, each containing at least one of the binding site amino acids, to find its position in the reference sequence by simple string search. From this mapping procedure we could define a continuous sequence region containing all the binding site residues.

For those human target proteins having several binding site lists (originated from different protein–ligand complexes), we clustered and aligned the obtained sequence regions and manually revised each cluster. From this analysis we defined a unique consensus binding region sequence for each target protein.

#### 3.1.4. Searching for Similar Binding Sites in Fungal Proteomes

The binding region sequences for the obtained set of human targets, as defined in the previous step, were used as query sequences for BLAST searches [50] in fungal proteomes, aiming to focus the search into regions that are more likely to be conserved among evolutionary distant organisms, such as humans and fungi. For comparison purposes, we performed BLAST searches using also the full sequences of the human targets.

For the subsequent analyses, we considered as hits only those alignments covering > 80% of the query sequence (i.e., the binding region sequence). The obtained alignments were then used to establish functional relationships between the binding pocket residues of the human targets and the corresponding amino acids in the fungal sequences. This way, the fungal binding sites became also defined at the amino acid level, as illustrated in Figure 7. The similarity (percent of amino acid identity) between a human binding site and its corresponding fungal binding pocket was evaluated taking into account only the binding pocket residues. Lastly, we analyzed the alignments showing > 70% identity for the set of binding pocket residues. From the DrugBank we retrieved the list of approved drugs for a small set of these human targets, using also web services such as Drugs.com (“Drugs.Com | Prescription Drug Information, Interactions & Side Effects,” 2021, last accessed on 10 April 2022).

### 3.2. Fungal Proteomes Included in the Study

We analyzed the proteomes of six fungal species: *Aspergillus fumigatus* (UP000002530), *Candida albicans* (UP000000559), *Candida parapsilosis* (UP000005221), *Cryptococcus neoformans* (UP000002149), *Fusarium oxysporum* (UP000009097) and *Histoplasma capsulatum* (UP000008142), retrieved from the UniProt database.

### 3.3. Homology Modeling and Molecular Docking

For homology modeling of fungal proteins, we used the SwissModel server [51]. Structural models of the *Histoplasma capsulatum* protein with UniProt identifier F0UAN5 and the *Fusarium oxysporum* protein A0D2XNJ1 were constructed using as template the crystal structure of human MEK1 in complex with an inhibitor (PDB code 3dv3) [52]. AutoDock Tools [53] was employed to prepare molecules for docking simulations, which were carried out with AutoDock Vina [54] using default parameters and a box enclosing the non-ATP competitive binding site.

### 3.4. In Vitro Assays of MEK Inhibitors

The in vitro tests to assess the susceptibility to MEK inhibitors were carried out in 96-well microplates, seeding 300,000 cells/well for yeasts (*Histoplasma capsulatum*, *Cryptococcus neoformans*, *Candida albicans* and *Candida parapsilosis*) and 40,000 conidia/well for *Fusarium oxysporum* and *Aspergillus fumigatus*. *Histoplasma capsulatum* was cultured for 6 days in HAMF12 medium supplemented with cysteine and glutamine. The other yeasts were cultured in RPMI 1640 supplemented with 2% glucose for 24 h (for the two *Candidas*) or 72 h (*Cryptococcus*), all of them at 37 °C and stirring at 150 rpm.

MEK inhibitors were purchased from Cayman Chemicals (Ann Arbor, MI, USA). For each compound, the maximum tested concentration was determined by the solubility data reported by the manufacturer. Each inhibitor was dissolved either in DMSO or ethanol according to manufacturer’s instructions. The stock solution for each compound was used at 1% as maximum, so that the DMSO concentration in the culture medium (kept at 1%) would not have toxic effects on the fungi. The compounds were tested also with the addition of 0.002% SDS, which most likely increased their solubility. Controls with 1% DMSO or ethanol, alone or combined with 0.002% SDS, were included in each microplate. To determine the half maximal inhibitory concentration (IC50), a 2-fold dilution series of 4 or 5 inhibitor concentrations was used. Fungal viability was determined using the XTT colorimetric assay.

## 4. Concluding Remarks

We have developed a strategy for a rational, structure-based approach to drug repositioning and new drug design, which can be applied not only to infectious fungi, but also to other pathogens. Following this methodology, we have identified fungal proteins having high binding site similarities with human targets of drugs that have shown inhibitory effects in fungi. These results not only support the developed strategy, but also contribute to identify the fungal targets responsible for these effects. Importantly, they also expose new routes to explore many drugs and inhibitors not yet tested in fungi.

Not all the identified fungal proteins, even if they are essential for the microorganism, are suitable for drug repositioning to treat fungal infections, especially in cases where the treatment produces severe side effects (as for many cancer drugs) or when it has immunosuppressive effects, which opens a door to opportunistic mycotic and bacterial infections. For a number of human targets, however, the available drugs may have only mild secondary effects, so they might be used to treat fungal infections if they show strong cytotoxic effects on these pathogens. Last but not least, the small structural differences in binding pocket architecture between some pairs of human and fungal proteins can be exploited to design specific antifungal drugs.

## Figures and Tables

**Figure 1 molecules-28-00692-f001:**
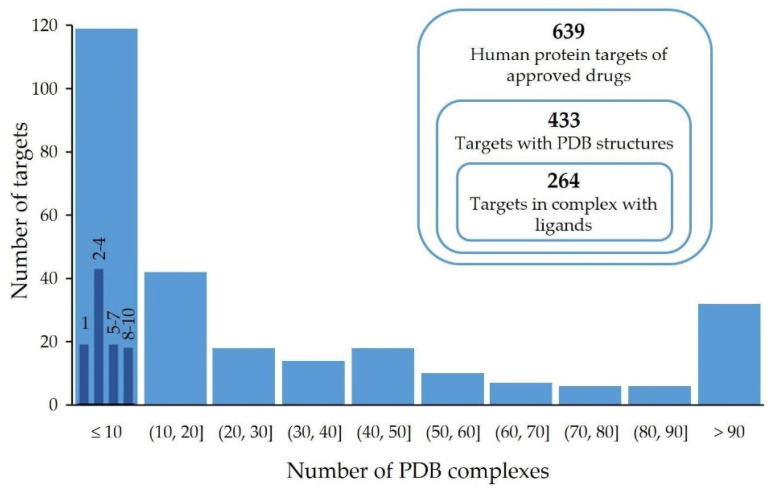
Distribution of the number of human targets per number of PDB protein–ligand complexes in the analyzed set of 264 proteins from the pharmacolome. For the first gross interval [1, 10), a distribution in smaller intervals (1, 2–4, 5–7 and 8–10) is shown in dark blue bars.

**Figure 2 molecules-28-00692-f002:**
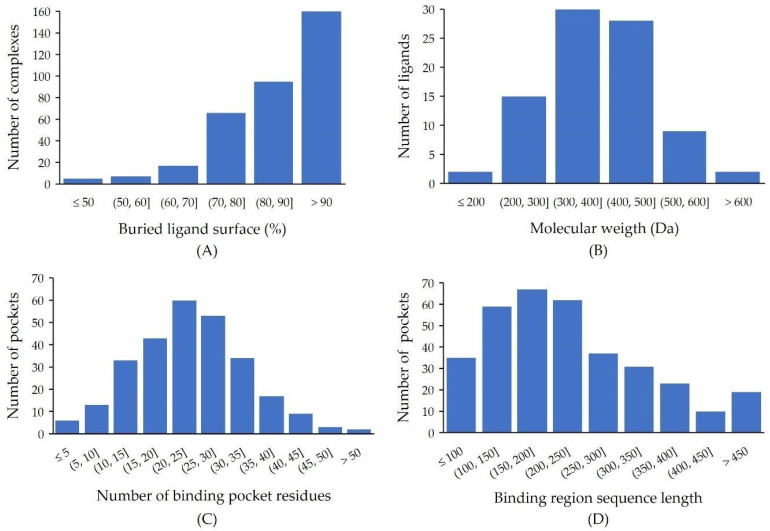
Statistics from the protein–ligand complexes of targets from the human pharmacolome. (**A**) Histogram of the buried ligand area upon complexation for FDA-approved drugs (data for cholic acid were omitted). (**B**) Distribution of ligand molecular weights for FDA-approved drugs. (**C**) Distribution of the number of residues per binding pocket for the 264 selected human targets. (**D**) Distribution of binding region sequence lengths for the 264 human targets.

**Figure 3 molecules-28-00692-f003:**
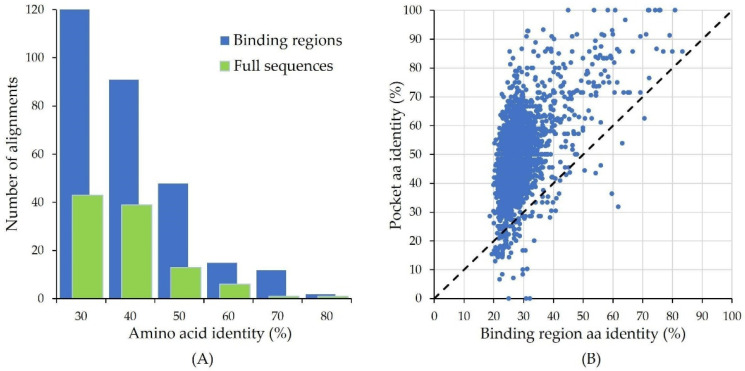
(**A**) Distribution of aa identity percent in the BLAST sequence alignments obtained for the *Histoplasma capsulatum* proteome, using as input either the full sequences (green bars) or the binding region sequences (blue bars) for the selected 264 proteins from the human pharmacolome. (**B**) Scatter plot showing the higher similarity between human and fungal binding pockets, as compared with the similarity between the binding region sequences that encompass the binding pocket amino acids.

**Figure 4 molecules-28-00692-f004:**
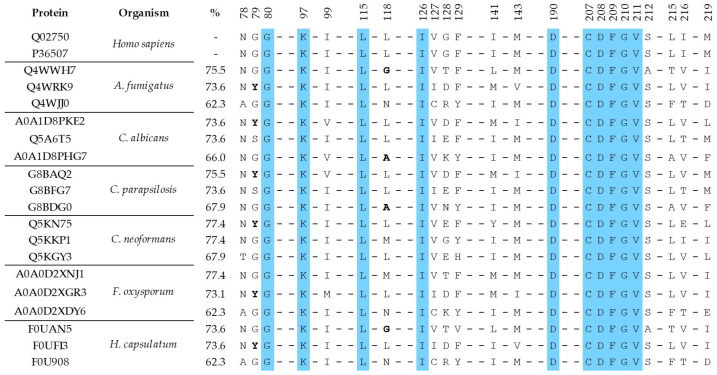
Alignment of the binding pocket residues for the non-ATP competitive site of MEK1/2 (Q02750 and P36507), with the corresponding residues in the identified fungal proteins having similar binding pockets, for the six analyzed species. The third column in the table shows the calculated aa identity percentages for the set of binding pocket residues. Fully conserved positions are highlighted in blue; non-conservative substitutions at positions 79 and 118 are marked in bold.

**Figure 5 molecules-28-00692-f005:**
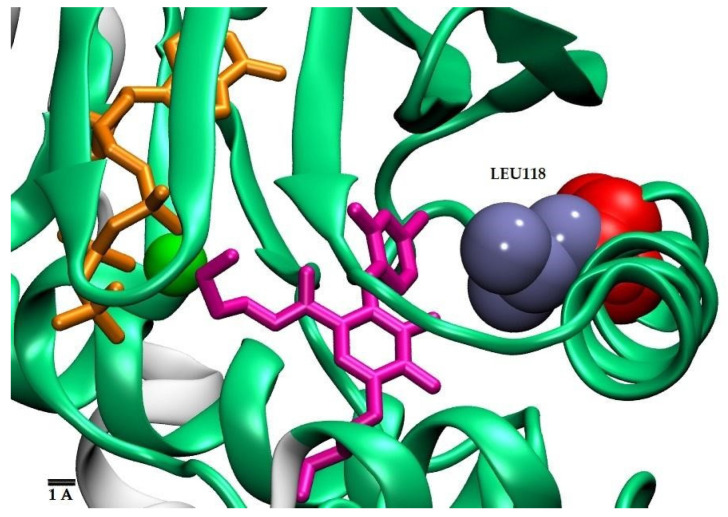
Structure of the MEK1/2 binding site showing the Leu 118 side chain (in blue spheres). Mutation of this residue by Glycine (in red spheres) creates a void space that can be occupied by atoms of fitting compounds.

**Figure 6 molecules-28-00692-f006:**
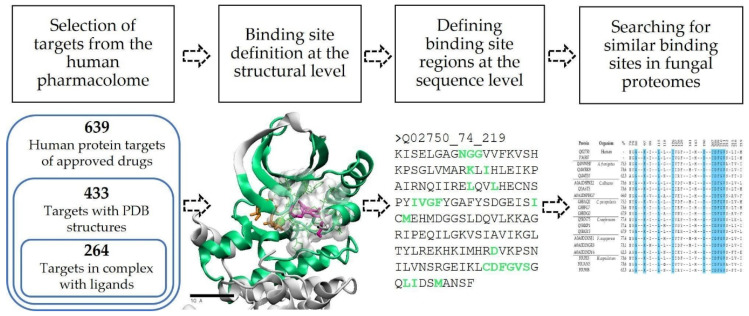
Computational strategy to search for fungal proteins with similar binding sites, taking as reference a selected set of proteins from the human pharmacolome.

**Figure 7 molecules-28-00692-f007:**
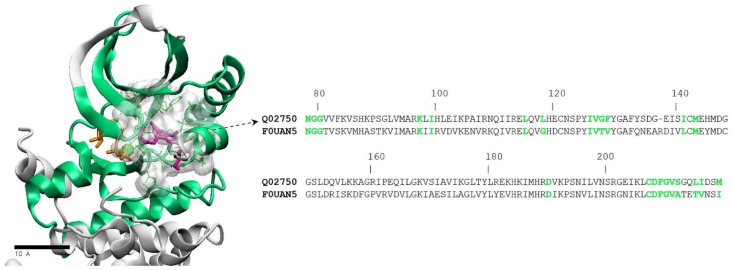
Definition of the binding site pocket and binding region sequence for the human MEK1 target and a fungal protein from *Histoplasma capsulatum* having a highly similar region. The continuous binding region sequence is represented as a green ribbon in the structure (PDB code 3dv3) and shown in full in one-letter code. Binding pocket amino acids are shown with their side chains (green, thin sticks) enclosed in a whitish volume, and are highlighted in green bold letters in the sequence. The MEK1 inhibitor in the 3dv3 structure is shown in thick sticks, colored in magenta. The ATP ligand is shown in orange sticks.

**Table 1 molecules-28-00692-t001:** Proteins with similar binding pockets (≥70% aa identity) for the six fungal species.

Human Protein Target	UniProt Code	Fungal Species *											
		*Histoplasma capsulatum*	%	*Aspergillus fumigatus*	*%*	*Candida albicans*	%	*Candida parapsilosis*	%	*Cryptococcus neoformans*	%	*Fusarium oxysporum*	%
Ribonucleotide reductase	P23921	F0U515	100	Q4WNR5	100	Q5A0N3	96	G8BE28	96	Q5KGK1	100		
Ornithine decarboxylase	P11926	F0U9K9	100	Q4WBW7	95	P78599	100	G8B654	100	Q5KJY8	100	A0A0D2XUF0	95
HMG-CoA reductase	P04035	F0UKH1	97	Q4WHZ1	97	A0A1D8PD39	100	G8B666	100	Q5KEN6	93	A0A0D2XPE5	97
P-glycoprotein 1	P08183	F0UGI7	93	Q4WTT9	87					Q5KN49	87	A0A0D2XVG0	80
IMP dehydrogenase 2	P12268	F0UKI1	93	Q4WHZ9	93	Q59Q46	97	G8B7I7	97	Q5KP44	97	A0A0D2XAA8	93
FPP synthase	P14324	F0UP55	91	Q4WEB8	91	A0A1D8PH78	91	G8B7C3	91	Q5KG83	96	A0A0D2XDQ9	91
Ribonucleotide reductase	P23921	F0U515	91	Q4WNR5	87	Q5A0N3	96	G8BE28	91	Q5KGK1	91	A0A0D2XVB2	91
Glutathione reductase	P00390	F0UFI2	91	Q4WRK8	91	Q6FRV2	73			Q5KH19	91	A0A0C4DHU2	91
Xanthine dehydrogenase	P47989	F0UCF6	90	Q4WQ15	90							A0A0D2Y4E4	90
DNA topoisomerase 2-β	Q02880	F0UWA9	87.5	Q4WLF7	75	O93794	82			Q5KP97	71	A0A0C4DHU5	87
Histone deacetylase 2	Q92769	F0UKC3	87	Q4WHY0	87	Q5ADP0	95.7	G8BBB0	96	Q5KF65	95.7	A0A0D2X821	87
GSK-3 beta	P49841	F0UQX6	87	Q4WDL1	93	A0A2H0ZU47	83	G8BDX2	80	Q5KMR8	90	A0A0D2XCF2	87
DHOdehase	Q02127	F0UDX1	87	Q4X169	87	Q874I4	100	G8BA68	100	Q5KK62	100		
Histone deacetylase 7	Q8WUI4	F0UVW7	85	Q4WE71	75	Q5A960	80	G8BBK1	80	Q5KL48	80	A0A0D2YC83	72.7
Thymidylate synthase	P04818	F0URV8	84	Q4W9N9	88	A0A2H0ZCG7	88	G8BCL7	88	P0CS12	78	A0A0D2X8Z9	88
PKC-alpha	P17252	F0UE28	84	Q4WVG0	84	A0A2H1A7H5	84	G8BAI0	79	A0A0S2LIC5	84	A0A0D2XWP4	89.5
IMP dehydrogenase 2	P12268	F0UKI1	82	Q4WHZ9	86	Q59Q46	82	G8B7I7	86	Q5KP44	82	A0A0D2XAA8	82
IMPase 1	P29218	F0UHX4	82	Q4WEX3	82	Q6FSE7	87			Q5KKG2	83	A0A0D2XJP5	78
FPP synthase	P14324	F0UP55	82	Q4WEB8	82	A0A1D8PH78	77	G8B7C3	77	Q5KG83	74	A0A0D2XDQ9	82
GARS/AIRS/GART	P22102	F0UHE5	81	Q4WDH1	81	A0A1D8PE67	75	G8BD50	75	Q5K7B4	75	A0A0D2Y2N6	88
Histone deacetylase 6	Q9UBN7	F0UVW7	81.2			Q5A960	75	G8BBK1	75	Q5KL48	86.7	A0A0D2YC83	87.5
DHOdehase	Q02127	F0UDX1	80	Q4X169	88								
Histone deacetylase 8	Q9BY41	F0UKC3	80	Q4WHY0	80	A0A2H0ZKW1	80	G8BBB0	76	Q5KF65	76.2	A0A0D2X821	80
PPIase FKBP1A	P62942	F0URT3	78			P28870	74			P0CP94	78		
HMG-CoA reductase	P04035	F0UKH1	77	Q4WSY2	77	A0A1D8PD39	77	G8B666	77	Q5KEN6	85	A0A0D2XPE5	77
Histone deacetylase 1	Q13547	F0UKK7	77	Q4WI19	77	A0A1D8PSA6	82	G8BBQ5	88	Q5KF65	71	A0A0D2X821	71
Rho kinase 2	O75116	F0UBW5	76	Q4WQ81	76	Q5AP53	73	G8BKE8	71	Q5KEJ1	73		
ALDH class 2	P05091	F0UNE9	75	Q4WM26	79	Q6FPK0	74			Q5KEX3	73	A0A0D2XAL2	74
CPSase 1	P31327	F0UNF7	75										
Tubulin beta-3 chain	Q13509	F0UQK5	74	Q4WA70	93	A0A1D8PC97	96	G8B7W7	96	Q5K876	89	A0A0C4DHQ2	93
MEK2	P36507	F0UAN5	74	Q4WWH7	76	Q6FQU4	82	G8BFG7	79	Q5KKP1	87	A0A0D2XNJ1	79
MEK1	Q02750	F0UAN5	74	Q4WWH7	76	A0A2H0ZYL6	77	G8BAQ2	76	Q5KKP1	77	A0A0D2XNJ1	77
Succinate dehydrogenase	P51649	F0U4T1	72.2	Q4WPA5	78	Q6FVP8	81	G8B862	78	Q5K8N2	81	A0A0D2Y168	88
CFTR	P13569	F0ULL9	72	Q4WIK7	74	Q6FWS5	79			Q5KL35	78		
DPDE4	Q07343	F0UHN7	71.4										
FADK 2	Q14289	F0UKJ3	70.8	Q4X028	76	A0A2H0ZKT3	83					A0A0D2XB54	71
MAPK 11	Q15759	F0USW8	70,4	Q4WSF6	70	Q92207	70	G8BE43	70	Q5KC34	70	A0A0D2XQS0	74
ATP-binding cassette G2	Q9UNQ0	F0U6H3	70	Q4WXJ0	70	Q6FQ96	76.2			Q5KCK1	71.4	A0A0D2Y998	71.4
IDPc	O75874			Q4WX92	100	A0A1D8PHH7	100	G8BDJ9	100	Q5KLU0	100	A0A0D2XM79	100
ICD-M	P48735			Q4WX92	100	A0A1D8PS79	100	G8BDJ9	100	Q5KLU0	100	A0A0D2XM79	100
DNA topoisomerase 2	P11388			Q4WLF7	89	A0A1D8PMM1	96	G8BHF5	96	Q5KP97	86	A0A0C4DHU5	93
PARP-2	Q9UGN5			Q4WU62	83.9							A0A0D2XUC2	87.1
IDPc	O75874			Q4WX92	77	A0A2H0ZLU3	77					A0A0D2XM79	71
Cyclin-dependent kinase 6	Q00534			Q4WN13	77	P43063	73	G8BG79	73			A0A0D2Y7P8	71
Thioredoxin reductase 1	Q16881			Q4WRK8	73.9								
DHOdehase	Q02127					Q874I4	84	G8BA68	84	Q5KK62	74		
Tyrosine kinase CSK	P41240					A0A1D8PR87	80	G8BKZ2	80				
ROCK-I	Q13464					Q6FP74	71			Q5KEJ1	70		
BCNG-2	Q9UL51					Q59V20	70,6						
Proto-oncogene c-Src	P12931					Q9Y7W4	70						
NTK38	P51813											A0A0C4DJR2	73
CFTR	P13569											A0A0D2XXA6	71
ALDH class 2	P05091											A0A0D2YFW3	70

* For each human target and each proteome, only the fungal protein with the highest aa identity percent is shown. The color code goes from dark to light gray following the decreasing percent of pocket aa identity (from 100% to 70%).

**Table 2 molecules-28-00692-t002:** Result of the in vitro susceptibility assays (IC50 values, (µM)).

Inhibitor	Fungal Species											
	*Histoplasma capsulatum*		*Cryptococcus neoformans*		*Candida albicans*		*Candida parapsilosis*		*Fusarium oxysporum*		*Aspergillus fumigatus*	
	Solvent *	+SDS **	Solvent	+SDS	Solvent	+SDS	Solvent	+SDS	Solvent	+SDS	Solvent	+SDS
cobimetinib	53	<12.5	>188	>188	>188	>188	>188	158	~100	65	>100	82
GDC-0623	114	63	194	83	>219	>219	>219	>219	>100	~100	8	7
myricetin	>251	36	>251	>251	>251	>251	>251	>251	>251	>251	>251	>251
TAK-733	>198	198	>198	198	>198	198	>198	>198	>100	>100	>100	54
AZD-6244	>328	>328	>328	>328	>328	>328	>328	>328	>328	>328	>328	>328
refametinib	25	<17.5	81	54	175	114	>175	175	ND	ND	ND	ND
trametinib	>49	>49	>49	>49	>49	>49	>49	>49	>49	>49	>49	>49

IC50 values < 100 µM are marked in bold and shadowed in gray. The “<” and “>” signs are used when the IC50 value is lower/greater than the minimum/maximum tested concentration. * Compounds were dissolved in DMSO or ethanol, and added to culture medium. ** Same as above, with the addition of 0.002% SDS.

## Data Availability

The data presented in this study are contained in the article tables and Appendix A.

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
