# Peer review of "Identifying Potential Molecular Targets in Fungi Based on (Dis)Similarities in Binding Site Architecture with Proteins of the Human Pharmacolome"

_molecules, 2023, doi:10.3390/molecules28020692_

Round 1

Reviewer 1 Report

In this manuscript, the authors presented a computational workflow to identify potential druggable targets in Fungi. The systematic computational studies that integrate binding site sequence similarity search, docking, as well as experimental validation was impressive. However, the presented computational workflow has significant flaws in theory and designs, which didn't show practical potentials in related field. 

1). It is inaccurate to use human protein sequences to search similar Fungi proteins. The methodology doesn't seem sound to me. 

2) The "similar" Fungi proteins found by similarity search against human pharmacolome doesn't means that those proteins are potential molecular targets. There is no pharmacological support for this approach. 

Given those, I will suggest the rejection of this paper.

Author Response

With all due respect, we disagree with the evaluation given by Reviewer 1. Below are our responses to the two presented points.

Point 1. It is inaccurate to use human protein sequences to search similar Fungi proteins. The methodology doesn't seem sound to me. 

Response:

This comment , rather than a criticism, reflects a personal opinion we disagree with. In fact, our approach has roots in the recent scientific literature, particularly in the field of drug repurposing.

Sequence comparison between human and fungal proteins is made, for example in the paper by Santos et al. (2017), published in Nature Reviews Drug Discovery, which was an important source of information for our work (Reference 25). References 15 and 16 in our manuscript also discuss this kind of procedure.

Here are a few additional examples where sequence similarity between human and fungal/protozoan proteins is used to identify potential targets for drug repurposing:

- Cheng et al. Prediction of drug-target interactions and drug repositioning via network-based inference. PLoS Comput Biol. 8:e1002503, 2012.
- Sateriale et al. Drug repurposing: mining protozoan proteomes for targets of known bioactive compounds. J Am Med Inform Assoc. 21:238-44, 2014.
- de Oliveira et al. Drug Repurposing for Paracoccidioidomycosis Through a Computational Chemogenomics Framework. Front Microbiol. 10:1301, 2019.
- Verma et al. Target-based drug repurposing against Candida albicans-A computational modeling, docking, and molecular dynamic simulations study. J Cell Biochem. 123:289-305, 2022.

We copy in here an excerpt from a recent review published in Nature Microbiology (Farha & Brown. Drug repurposing for antimicrobial discovery. Nat Microbiol 4, 565–577, 2019), which explicitly supports our bioinformatics approach of comparing human and fungal sequences for drug repurposing:

"Interestingly, fungi and parasites, for example, have targets that are more similar to humans. Indeed, eukaryotic microorganisms are more similar to their hosts than prokaryotic pathogens in terms of their biochemistry and metabolism, genetic composition, cell architecture and biology. Thus it is tempting to assume that underlying polypharmacology —35% of active compounds are thought to hit more than one target — has helped enable repurposing opportunities for eukaryotic microbial pathogens, more so than prokaryotic pathogens."

Finally, we provide in the manuscript a sound experimental evidence of the validity of our approach, by successfully probing in various fungi a group of predicted inhibitors of the human MEK protein.

Point 2. The "similar" Fungi proteins found by similarity search against human pharmacolome doesn't means that those proteins are potential molecular targets. There is no pharmacological support for this approach.

Response:

Perhaps this criticism is due to different interpretations of the adjective "potential" by Reviewer 1 and us. According to various dictionaries (Britannica, Cambridge, Merriam-Webster), this adjective means: "possible but not yet achieved", "having a possibility of happening or being", "expressing possibility".

This is the meaning we have in mind here. Our bioinformatics approach predicts "potential" (possible) targets. Whether or not they can become actual targets in fungi, is something that has to be supported experimentally, as Reviewer 1 points out. We do not see any contradiction here.

In fact, even for the predicted fungal analogs of the human MEK protein, where we obtained remarkable results in the growth inhibition experiments, we were cautious in our conclusions, noting that we cannot yet declare any of these proteins as a valid pharmacological target in fungi, since "reliable target validation would require complementary experiments, e.g., genetic manipulations to affect protein expression." (page 12, last para).

Reviewer 2 Report

Bedoya-Cardona and Colleagues developed in their manuscript a structure-based strategy to drug repurposing applicable to infectious fungi and other pathogens. They used structural information from thousands of human protein target-inhibitor complexes identifying several proteins in five fungal species. They also presented an in vitro case study with a set of selected inhibitors of the human mito-gen-activated protein kinases 1/2 (MEK1/2), several of which showed a marked cytotoxic activity in different fungal species.    The manuscript, I think, is suitable for publication after typos check since it is sounding and very well written. The design in appropriate and all the in silico procedures are easily applicable.    All the best.

Author Response

We thank Reviewer 2 for his/her encouraging comments on our manuscript.

Reviewer 3 Report

It is an interesting and readable bioinformatic paper which suggest a new strategy: beside drug repositioning i.e., using approved drugs or compounds that have shown an acceptable safety profile in clinical trials as antifungal agents, slight modification of these drugs based on the differences of the binding pockets of the human and fungal proteins may lead new effective antifungal molecules.

I guess that the paper can be publish almost as is except References. I have never come across such an incomplete bibliography.

Minor suggestion:

Line 343: Delete Cryptosporidium parvum (and Ref 43). It is an apicomplexan parasite not a fungus.

References:

Use either an abbreviated or full journal title consistently!

Do not enter the name of the publishers! (e.g., W.B. Saunders, MDPI AG, Elsevier, etc.)

In all cases, enter the page numbers or the article ID! Sometimes other data (volume, issue) are also missing.

Some examples:

Ref 1 – Add: Article number: 404

Ref 11 - pp. 673-683. 

Ref 17 – Paper 488

Ref 18 – pp. 5855-5862

Ref 19 – vol. 92, no. 4, pp. 368-373.

Ref 23 – vol. 62, no. 10, e01084-18.

I'm tired of correcting here. Please, complete the further missing volume and page numbers of the references. Only one more example. Ref 25 – you did not give any data. (Volume 20, Issue 7, Page: 509 - 516

Author Response

We thank Reviewer 3 for his/her positive comments on our manuscript. 

Minor suggestion:

Line 343: Delete Cryptosporidium parvum (and Ref 43). It is an apicomplexan parasite not a fungus.

Response/ We have eliminated this incorrect reference.

Regarding the "References" section, we acknowledge the inadequacies in the Reference list, which we have now carefully checked and corrected, using the MDPI template.

Thanks again to Reviewer 3 for such detailed revision of our manuscript.